# Clinical Validation of the Fully Automated NeuMoDx HPV Assay for Cervical Cancer Screening

**DOI:** 10.3390/v14050893

**Published:** 2022-04-25

**Authors:** Daniëlle A. M. Heideman, Anja Oštrbenk Valenčak, Saskia Doorn, Jesper Bonde, Peter Hillemanns, Grega Gimpelj Domjanič, Jana Mlakar, Albertus T. Hesselink, Chris J. L. M. Meijer, Mario Poljak

**Affiliations:** 1Department of Pathology, Amsterdam UMC Location Vrije Universiteit Amsterdam, De Boelelaan 1117, 1081 Amsterdam, The Netherlands; dam.heideman@amsterdamumc.nl (D.A.M.H.); cjlm.meijer@amsterdamumc.nl (C.J.L.M.M.); 2Cancer Center Amsterdam, Imaging and Biomarkers, 1081 Amsterdam, The Netherlands; 3Institute of Microbiology and Immunology, Faculty of Medicine, University of Ljubljana, 1000 Ljubljana, Slovenia; anja.ostrbenk@mf.uni-lj.si (A.O.V.); grega.gimpelj-domjanic@mf.uni-lj.si (G.G.D.); jana.mlakar@mf.uni-lj.si (J.M.); 4Self-Screen, 1081 Amsterdam, The Netherlands; s.doorn@self-screen.nl (S.D.); at.hesselink@self-screen.nl (A.T.H.); 5Molecular Pathology Laboratory, Department of Pathology, Hvidovre Hospital, 2650 Hvidovre, Denmark; jesper.hansen.bonde@regionh.dk; 6Department of Gynecology and Obstetrics, Hannover Medical School, 30625 Hannover, Germany; hillemanns.peter@mh-hannover.de

**Keywords:** human papillomavirus, NeuMoDx HPV Test Strip, clinical performance, reproducibility, cervical cancer screening, liquid medium

## Abstract

The NeuMoDx HPV assay is a novel fully automated, real-time PCR-based assay for the qualitative detection of high-risk human papillomavirus (HPV) DNA in cervical specimens. The assay specifically identifies HPV16 and HPV18 and concurrently detects 13 other high-risk HPV types at clinically relevant infection levels. Following the international guidelines, the clinical performance of the NeuMoDx HPV assay for cervical intraepithelial neoplasia grade 2 or worse (CIN2+) against the reference standard Hybrid Capture 2, as well as intra- and inter-laboratory reproducibility were assessed on PreservCyt samples. The clinical accuracy of the assay was additionally evaluated against the clinically validated Alinity m HR HPV and COBAS 4800 HPV Test on PreservCyt samples, and against the clinically validated HPV-Risk assay on SurePath samples. The NeuMoDx HPV assay performance for CIN2+ was non-inferior to the reference methods on both sample types (all *p* < 0.05), and showed excellent intra- and inter-laboratory reproducibility (95.7%; 95% CI: 93.9–97.3; kappa value 0.90 (95% CI: 0.86–0.94); and 94.5%; 95% CI: 92.6–96.2; kappa value 0.87 (95% CI: 0.82–0.92), respectively). In conclusion, the NeuMoDx HPV assay meets international guideline criteria for cross-sectional accuracy and reproducibility, and performs equally well on cervical screening specimens collected in two widely used collection media. The NeuMoDx HPV assay fulfils the requirements to be used for primary cervical screening.

## 1. Introduction

Worldwide, an increasing number of cervical cancer screening programs are converting from cytology to primary human papillomavirus (HPV)-based screening because of the better clinical performance for the detection of cervical cancer and its high-grade precursor lesions [1,2]. HPV assays need to be clinically validated and comply with the international guidelines for HPV DNA test requirements to assure a high quality of primary HPV-based cervical screening [3]. In addition, the operational aspects of HPV assays, such as the reproducibility of results and a fully automated workflow, are important factors for the execution of the screening program. The COVID-19 pandemic further stresses the need for laboratory systems to be flexible, economical on resources and reagents, and to run a large portfolio of assays in order not to compromise other molecular diagnostics and healthcare. The NeuMoDx Systems (NeuMoDx 96 and NeuMoDx 288; NeuMoDx Molecular, a QIAGEN company, Ann Arbor, MI, USA) are such sample-to-result fully automated, real-time PCR-based solutions capable of running 20–30 different assays simultaneously. The systems integrate reagent storage, specimen preparation, nucleic acid extraction, PCR setup, amplification, and detection, as well as results analysis and reporting. With true-random access and cartridge-based sample preparation and PCR, the assay results can be available in 60–80 min, depending on the assay type (DNA or RNA). All reagents are stable at ambient conditions as well as on the worktable of the NeuMoDx Molecular System. The recently developed NeuMoDx HPV Test Strip utilizes the design of the HPV-Risk assay (self-screen B.V.) for the qualitative detection of HPV DNA in cervical specimens [4] with transition to the stabilised reagent format PCR master mix for implementation on the NeuMoDx Systems. Its predecessor, the HPV-Risk assay, is clinically validated for cervical cancer screening purpose with both cervical screening samples collected in PreservCyt and SurePath, as well as for self-collected cervico-vaginal specimens [4,5,6]. The assay targets the E7 region of 15 hrHPV types (HPV16, 18, 31, 33, 35, 39, 45, 51, 52, 56, 58, 59, 66, 67, and 68). In contrast to the L1 region, the E7 region is retained in virtually all cervical cancers. Thereby, E7 as a target shields against potential false-negative results. The assay confers partial genotyping by individually reporting HPV16 and HPV18, and concurrently detects the 13 other common high-risk types as a pool.

Here, the NeuMoDx HPV assay, i.e., NeuMoDx HPV Test Strip implemented on the NeuMoDx Molecular Systems, was evaluated according to the international guidelines for HPV DNA test requirements for cervical screening in women 30 years and older [3]. The NeuMoDx HPV assay was clinically evaluated for the detection of cervical intraepithelial neoplasia (CIN) grade 2 or worse (CIN2+) in comparison with the reference standard test Hybrid Capture 2 (HC2; digene HC2 HPV DNA Test, QIAGEN, Gaithersburg, MD, USA) on 895 PreservCyt (Hologic Inc., Marlborough, MA, USA) samples and intra-/inter-laboratory reproducibility was assessed. Additionally, the results of the NeuMoDx HPV assay were compared with those of two other clinically validated tests, the COBAS 4800 HPV (Roche Molecular Systems, Pleasanton, CA, USA) and Alinity m HR HPV assay (Abbott Molecular, Des Plaines, IL, USA), both performed on the same collection of PreservCyt samples. The clinical performance of the NeuMoDx HPV assay was further evaluated on 948 SurePath (Becton Dickinson, Sparks, MD, USA) samples against the clinically validated HPV-Risk assay.

## 2. Materials and Methods

### 2.1. NeuMoDx HPV Assay

The NeuMoDx HPV assay (NeuMoDx HPV Test Strip, NeuMoDx molecular, a QIAGEN company, Ann Arbor, MI, USA) combines automated DNA extraction and amplification/detection by real-time PCR. The assay is designed for the qualitative detection of 15 high-risk HPV types at clinically relevant viral load levels in cervical specimens. The detection of target region in the E7 gene is by hydrolysis probes with 3 different fluorescent dyes, each representing different (pools of) targets, in particular HPV16, HPV18, and 13 other high-risk HPV types combined. The human β-globin gene is detected in a fourth channel using a probe labeled with a different fluorescent dye and serves as a qualitative endogenous sample process control to monitor for poor sampling, presence of inhibitory substances and/or for system, process, or reagent failures. The primers and probes together with the other elements necessary for amplification of the HPV and β-globin targets (e.g., Taq polymerase, salts, etc.) are dried using a proprietary process at the bottom of each well in a test strip and sealed from exposure to ambient light and atmosphere. All testing was performed according to the instructions of the manufacturer using the NeuMoDx 96 Molecular System (N96). Cervical specimens collected in PreservCyt collection medium) can be directly processed on the NeuMoDx system without sample pre-treatment utilising 0.25% of the original sample (0.05 mL). Specimens in SurePath collection medium require a pre-treatment to overcome the formaldehyde-induced crosslinking of the DNA. For this, the original SurePath sample is diluted 1:1 in NeuMoDx Viral Lysis buffer (NeuMoDx Molecular) and incubated at 90 °C for 20 min after which the sample is ready for loading on the system utilising 1% of the original sample (0.1 mL). The results processing algorithm is captured in an assay definition file (ADF) and scoring of a sample is based on second derivative Ct calling, endpoint fluorescence and endpoint fluorescence ratio. In samples where no HPV target(s) are scored, the sample is considered HPV negative only when β-globin amplification is detected. Samples where no β-globin nor HPV target(s) are scored are reported as Unresolved indicating invalid result.

### 2.2. Clinical Specimens

The study was performed using residual clinician-collected cervical specimens from women aged 30 years and older collected in PreservCyt collection medium (group 1) or in SurePath collection medium (group 2). All cervical specimens had been collected with the Cervex brush or the Combi-brush collection device (both Rovers Medical Devices, Oss, The Netherlands). The work in this study with human-derived material was conducted under applicable international, EU and national rules and legislation/regulations, with approval by the local ethics committees where applicable (National Medical Ethics Committee of the Republic of Slovenia, consent numbers 83/11/09 and 109/08/12; the Danish Data Inspection Agency J. No. AHH-2017-024, I-Suite: 05356; and the Medical Ethics Committee of Amsterdam UMC, location VUmc, number U2017.02).

#### 2.2.1. Group 1 (PreservCyt Collection Medium)

This series comprised 895 historical clinician-collected cervical screening samples including 68 samples from women with histologically confirmed CIN2+ (18 CIN2 and 50 CIN3; median age 39, range 30–76 years) and 827 from women attending a national screening programme with two consecutive normal cytology results (3 year interval; median age 39, range 30- 65 years), that were prospectively enrolled within a nationwide network of 16 outpatient gynaecology services in Slovenia between December 2009 and August 2016. The exclusion criteria for control group were attendance for a gynaecological examination after an atypical/abnormal cytology result, history of CIN of any grade or treatment for cervical disease in the preceding years, hysterectomy, and menstruation or pregnancy at presentation [7,8]. The clinically validated HC2 test served as a reference assay. All samples were additionally tested with COBAS 4800 HPV and Alinity m HR HPV assay, in accordance with the manufacturer’s instructions [9]. All samples with discordant results between NeuMoDx and comparator assays were tested by Anyplex II HPV28 Detection (Seegene, Seoul, South Korea), in accordance with the manufacturer’s instructions.

For analysis of intra-laboratory reproducibility over time and inter-laboratory agreement, 512 archived clinician-collected cervical samples of which one third tested HPV positive were used. The NeuMoDx HPV assay was performed twice blinded on this sample series with an average 23 (range 12–78) days interval between testing using N96 System at the same laboratory (Amsterdam, The Netherlands; hereafter referred to as laboratory 1). A third analysis was performed in a blinded manner using a second N96 System located at another laboratory (Ljubljana, Slovenia; referred to as laboratory 2).

#### 2.2.2. Group 2 (SurePath Collection Medium)

This series comprised 948 historical clinician-collected cervical screening samples including 106 samples from women with histologically confirmed CIN2+ (i.e., 39 CIN2, 63 CIN3, and 4 SCC (median age 38, range 30–58 years) and 842 samples from women who were without evidence of CIN2+ in up to 2 years of follow-up monitoring and twice tested cytology normal prior to inclusion (median age 43, range 30–59 years) [10]. The clinically validated HPV-Risk assay served as reference assay in this group [6].

### 2.3. Data and Statistical Analyses

Testing of the NeuMoDx HPV assay was performed in a blind manner for any test result as well as cytological and histological outcomes, and data were correlated after-wards. The clinical sensitivity and specificity values for the NeuMoDx HPV assay were compared with those of the reference or comparator assay using a non-inferiority score test as described by Tang et al. [11], with a relative sensitivity threshold for CIN2+ of 90% and a relative specificity threshold for CIN2+ of 98% [3]. For the intra-laboratory reproducibility and inter-laboratory agreement analyses of the NeuMoDx HPV assay, the agreement and kappa values according to Cohen’s kappa statistics for samples with valid test results were determined. For acceptable results, the 95% lower confidence bounds of the intra-laboratory reproducibility and inter-laboratory agreement values should both be ≥87%, with kappa values of ≥0.5 [3]. Genotype agreement was determined for samples that were positive in all runs/laboratories. Concordant genotype findings were defined as complete agreement, compatible findings as having at least one genotype category in common, and discordant findings as no similarity between detected genotype categories. All statistical calculations were performed using SPSS (version 28) and *p*-values ≤0.05 were considered statistically significant.

## 3. Results

### 3.1. Clinical Accuracy and Assay Reproducibility According to the International Guidelines on PreservCyt Cervical Specimens (Group 1)

A series of 895 cervical screening samples collected in PreservCyt (68 from women with CIN2+ and 827 from control women without evidence of CIN2+) was tested, of which 97.0% (868/895) of samples gave valid test results with the NeuMoDx HPV assay. For samples with valid test results, the clinical sensitivity for CIN2+ of the NeuMoDx HPV assay was found to be 95.5% (64/67; 95% CI: 86.6–98.8%) and the clinical specificity for CIN2+ was determined to be 96.4% (772/801; 95% CI: 94.8–97.5). For comparison, these figures were 97.0% (65/67; 95% CI: 88.7–99.5) and 93.8% (751/801; 95% CI: 91.8–95.3), respectively, for the reference assay HC2 (Table 1).

The relative sensitivity and specificity values of the NeuMoDx HPV assay compared to the reference assay HC2 were 0.98 (95% CI: 0.92–1.05) and 1.03 (95% CI: 1.01–1.05), respectively. Both the clinical sensitivity and clinical specificity for CIN2+ of the NeuMoDx HPV assay were non-inferior to that of the reference assay (non-inferiority score test, *p* = 0.018 and 0.0001, respectively).

The clinical sensitivity for CIN3+ of the NeuMoDx HPV assay was found to be 95.9% (47/49; 95% CI: 84.9–99.3%) and the clinical specificity for CIN3+ was determined to be 94.4% (773/819; 95% CI: 92.5–95.8). For comparison, these figures were 95.9% (47/49; 95% CI: 84.9–99.3%) and 91.7% (751/819; 95% CI: 89.5–93.5), respectively, for the reference assay HC2.

The intra-laboratory reproducibility over time and inter-laboratory agreement of the NeuMoDx HPV assay was determined on 512 cervical specimens. A valid test result was obtained for 99.6% (510/512) and 99.8% (511/512) of the samples tested in the two runs performed in laboratory 1, and for 99.8% (511/512) of the samples tested in laboratory 2. For samples with valid test results in all runs/laboratories (*n* = 509), the intra-laboratory reproducibility over time was 95.7% (447/509; 95% CI: 93.9–97.1) with a kappa value of 0.90 (95% CI 0.86–0.94; Table 2). The inter-laboratory agreement (i.e., laboratory 1, run 1 and laboratory 2, Table 2) was 94.5% (481/509; 95% CI: 92.6–96.2) and the kappa was 0.87 (95% CI 0.82–0.92). In addition, both the intra- and inter-laboratory genotyping agreement were high (Table 3 and Table 4).

### 3.2. Clinical Accuracy of the NeuMoDx HPV Assay for SurePath Cervical Specimens (Group 2)

A series of 948 cervical screening samples collected in SurePath (106 from women with CIN2+ and 842 from women without evidence of CIN2+) was evaluated, of which 98.6% (935/948) of samples gave valid test results with the NeuMoDx HPV assay. For samples with valid test results, the clinical sensitivity for CIN2+ of the NeuMoDx HPV assay was determined to be 92.5% (98/106; 95% CI: 85.6–96.2) and the clinical specificity for CIN2+ was calculated as 93.5% (775/829; 95% CI: 91.6–95.0). For comparison, these figures were 92.5% (98/106; 95% CI: 85.6–96.2) and 91.9% (762/829; 95% CI: 89.9–93.6), respectively, using the HPV-Risk assay (Table 5). The relative sensitivity and specificity values of the NeuMoDx HPV assay compared to the reference assay were 1.00 and 1.02, respectively. Both the clinical sensitivity and clinical specificity for CIN2+ of the NeuMoDx HPV assay were non-inferior to that of the reference assay (non-inferiority score test, *p* = 0.0009 and <0.0001, respectively).

### 3.3. Clinical Performance for CIN2+ of NeuMoDx HPV Assay in Relation to Other HPV Tests for PreservCyt Cervical Specimens (Group 1)

The performance of the NeuMoDx HPV assay in group 1 samples was further evaluated in relation to the COBAS 4800 HPV and Alinity m HR HPV assay. The clinical sensitivity and specificity of NeuMoDx HPV assay were non-inferior to those of COBAS 4800 HPV (*p* = 0.011 and *p* = 0.006, respectively) and non-inferior to those of Alinity m HR HPV assay (*p* = 0.03 and *p* < 0.0001, respectively). For detailed results see Table 6 and Table 7.

## 4. Discussion

In this study, we report on the clinical performance of the novel, fully automated, NeuMoDx HPV assay. The assay meets the clinical accuracy and reproducibility criteria of the international guidelines for HPV DNA test requirements [3] when compared with the standard reference test HC2. In addition, the NeuMoDx HPV assay performance was non-inferior to those of three other clinically validated HPV assays: HPV-Risk assay, COBAS 4800 HPV and Alinity m HR HPV assay. The NeuMoDx HPV assay is compatible with both cervical specimens collected in the liquid-based cytology media PreservCyt and SurePath. Collectively, these data support that the NeuMoDx HPV assay can be considered clinically validated for HPV primary screening; however, further studies are needed to assess its performance for ASC-US triage and/or test of a cure.

There are many HPV detection assays commercially available (>250); however, only a small number demonstrated full compliance with requirements of the 2009 international guidelines for HPV DNA tests [12], and an even smaller subset is validated for both most widely-used collection media, PreservCyt and SurePath. Given the different chemical formulation of these two collection media, the international HPV community requests that all HPV assays need validation on both ThinPrep and SurePath rather than to assume similar performance across both based on an evaluation on a single collection media [13]. Pretreatment of clinical samples collected in SurePath is generally required for optimal performance of PCR-based techniques to overcome the formaldehyde-induced cross-linking effect [14]. The standardised pretreatment protocol for the NeuMoDx HPV assay yielded an equally high success rate compared to PreservCyt, indicating compatibility of the assay with these two most commonly used preservative media. A possible limitation of the study is the use of archived residual clinical specimens which may result in a higher number of invalid samples. Nonetheless, valid results were achieved for the large majority of cervical samples used in this study. Another limitation may be that our results could possibly be biased because no histological endpoint was obtained in a number of control women. However, it is unlikely that this would have a marked influence on the outcome of this study given that these women were reported with two consecutive normal cytology results at least three years apart.

Clinical samples with discrepant test results between the NeuMoDx HPV assay and the reference assays were primarily found in controls. Discrepant results were not related to a specific HPV target, but mostly due to signals just above or below threshold settings of the assays (data not shown). This is a well-known cause of discrepancies for samples without underlying clinically meaningful disease harboring low viral load levels just around the clinical assay cut-off [15,16].

The NeuMoDx HPV assay has several advantages for cervical cancer screening purposes, including: (1) clinically relevant detection of all high-risk HPV genotypes (E7 region) with additional partial genotyping of HPV16 and HPV18; (2) being a fully automated “sample to result” system with a time to first result of approximately 60 min which is significantly shorter in comparison to the great majority of other HPV tests on the market, including the comparators used in this study; (3) utilising ambient-condition-stable reagents capable of being stored on the system, thereby eliminating the need for manual preparations, and (4) possessing high-throughput and high intra-/inter-system reproducibility. Herein, the reproducibility of testing on the N96 System was shown to be high, both intra- and inter-laboratory. Part of the samples (*n* = 169; 59.7% HPV-positive) were also subjected to testing on the N288 System, with similarly high inter-laboratory agreement results (92.9%; 95% CI: 88–96; kappa value of 0.86 (95% CI 0.78–0.93); data not shown). The ability of the NeuMoDx Systems to run different assays in parallel, both combinations of other CE-IVD assays and homebrew or laboratory developed tests, with a true random access and continuous loading workflow, enhances their utility in molecular diagnostics laboratories.

In conclusion, the NeuMoDx HPV assay has a high sensitivity and a high specificity for the detection of CIN2+, independent of cervical sample collection medium. The fully automated NeuMoDx HPV assay fulfils the requirements of the international guidelines for an HPV DNA test to be used for primary cervical screening.

## Figures and Tables

**Table 1 viruses-14-00893-t001:** Clinical performance for CIN2+ of NeuMoDx HPV assay as compared to Hybrid Capture 2 reference test on PreservCyt collection medium (group 1).

		Reference Assay Hybrid Capture 2			Non-Inferiority Score Test (*p*-Value)
		Negative	Positive	Total	
**Women with ≤CIN1**					
**NeuMoDx HPV assay**	Negative	746	26 ^##^	772	0.0001
	Positive	5 ^#^	24	29	
	Total	751	50	801	
**Women with CIN2+**					
**NeuMoDx HPV assay**	Negative	1	2 **	3	0.018
	Positive	1 *	63	64	
	Total	2	65	67	

Results with Anyplex II HPV28 Detection: (*) HPV16 (*n* = 1 CIN3); (**) HPV52 (*n* = 1 CIN3); HPV53, non-targeted by NeuMoDx (*n* = 1 CIN2); (#) HPV-negative (*n* = 5); (##) HPV16 (*n* = 3), non-16/18 other hrHPV (*n* = 13), HPV non-targeted by NeuMoDx (*n* = 4), HPV negative (*n* = 6).

**Table 2 viruses-14-00893-t002:** Intra-laboratory reproducibility and inter-laboratory agreement analyses of the NeuMoDx HPV assay on cervical specimens.

Lab 1/Run 1	Lab 1/Run 2		Total	Lab 2		Total
	Negative	Positive		Negative	Positive	
Negative	335	13	348	342	6	348
Positive	9	152	161	22	139	161
**Total**	344	165	509	364	145	509
Agreement	95.7% (95% CI: 93.9–97.3)			94.5% (95% CI: 92.6–96.2)		
Kappa value	0.90 (95% CI: 0.86–0.94)			0.87 (95% CI: 0.82–0.92)		

**Table 3 viruses-14-00893-t003:** Intra-laboratory genotype agreement ^1^.

Lab 1/Run 1	Lab 1/Run 2								Total
	Negative	HPV16	HPV18	Other HPV	HPV16 and 18	HPV16 and Other	HPV18 and Other	HPV16, 18 and Other	
Negative	335	2	-	11	-	-	-	-	348
HPV16	-	46	-	-	-	1	-	-	47
HPV18	-	-	2	-	-	-	-	-	2
Other HPV	9	-	-	78	-	-	-	-	87
HPV16 and 18	-	-	1	-	3	-	-	-	4
HPV16 and other	-	1	-	-	-	18	-	-	19
HPV18 and other	-	-	-	-	-	-	1	-	1
HPV16, 18 and other	-	-	-	-	-	-	-	1	1
**Total**	344	49	3	89	3	19	1	1	509

^1^ for samples with positive results in both runs: identical genotype 149/152 (98.0%); compatible 3/152 (2.0%). Kappa 0.97 (95% CI: 0.93–1.00).

**Table 4 viruses-14-00893-t004:** Inter-laboratory genotype agreement ^1^.

Lab 1/Run 1	Lab 2								Total
	Negative	HPV16	HPV18	Other HPV	HPV16 and 18	HPV16 and Other	HPV18 and Other	HPV16, 18 and Other	
Negative	342	-	-	6	-	-	-	-	348
HPV16	-	47	-	-	-	-	-	-	47
HPV18	-	-	2	-	-	-	-	-	2
Other HPV	22	-	-	65	-	-	-	-	87
HPV16 and 18	-	-	1	-	2	-	-	1	4
HPV16 and other	-	3	-	-	-	16	-	-	19
HPV18 and other	-	-	-	-	-	-	1	-	1
HPV16, 18 and other	-	-	-	-	-	-	-	1	1
**Total**	364	50	3	71	2	16	1	2	509

^1^ for samples with positive results in both runs: identical genotype 134/139 (96.4%); compatible 5/139 (3.6%). Kappa 0.95 (95% CI 0.90–0.99).

**Table 5 viruses-14-00893-t005:** Clinical performance for CIN2+ of NeuMoDx HPV assay as compared to HPV-Risk assay on SurePath collection medium (group 2).

		Reference Assay HPV-Risk Assay			Non-Inferiority Score Test (*p*-Value)
		Negative	Positive	Total	
**Women with ≤CIN1**					
**NeuMoDx HPV assay**	Negative	756	19 ^##^	775	<0.0001
	Positive	6 ^^^	48	54	
	Total	762	67	829	
**Women with CIN2+**					
**NeuMoDx HPV assay**	Negative	7	1 *	8	0.0009
	Positive	1 ^#^	97	98	
	Total	8	98	106	

Genotype findings: (*) non-16/18 other hrHPV (*n* = 1); (#) non-16/18 other hrHPV (*n* = 1); (##) HPV16 (*n* = 2); non-16/18 other hrHPV (*n* = 15); HPV16 and non-16/18 other HPV (*n* = 2); (^^^) non-16/18 other hrHPV (*n* = 6).

**Table 6 viruses-14-00893-t006:** Clinical performance for CIN2+ of NeuMoDx HPV assay on PreservCyt collection medium (group 1)—Comparator COBAS 4800 HPV assay.

		Comparator Assay COBAS 4800 HPV Assay			Non-Inferiority Score Test (*p*-Value)
		Negative	Positive	Total	
**Women with ≤CIN1**					
**NeuMoDx HPV assay**	Negative	751	21 ^##^	772	0.006
	Positive	7 ^#^	22	29	
	Total	758	43	801	
**Women with CIN2+**					
**NeuMoDx HPV assay**	Negative	2	1 *	3	0.011
	Positive	-	64	64	
	Total	2	65	67	

Relative sensitivity: 0.98 (0.92–1.05); relative specificity: 1.02 (1.00–1.04). Results with Anyplex II HPV28 Detection: (*) HPV33 (*n* = 1 CIN3); (#) HPV67 (*n* = 1); HPV70 (*n* = 1); HPV negative (*n* = 5); (##) HPV16 (*n* = 3); HPV18 (*n* = 1); non-16/18 other hrHPV (*n* = 16); negative (*n* = 1).

**Table 7 viruses-14-00893-t007:** Clinical performance for CIN2+ of NeuMoDx HPV assay in women aged >30 years on PreservCyt collection medium (group 1)—Comparator Alinity m HR HPV assay.

		Comparator Assay Alinity HPV Assay			Non-Inferiority Score Test (*p*-Value)
		Negative	Positive	Total	
**Women with ≤CIN1**					
**NeuMoDx HPV assay**	Negative	743	29 ^##^	772	<0.0001
	Positive	6 ^#^	23	29	
	Total	749	52	801	
**Women with CIN2+**					
**NeuMoDx HPV assay**	Negative	1	2 *	3	0.03
	Positive	-	64	64	
	Total	1	66	67	

Relative sensitivity: 0.97 (0.91–1.03); relative specificity: 1.03 (1.01–1.05). Results with Anyplex II HPV28 Detection: (*) HPV33 (*n* = 1 CIN3); HPV52 (*n* = 1 CIN3) (#) HPV67 (*n* = 1); HPV negative (*n* = 5); (##) HPV16 (*n* = 3); HPV18 (*n* = 4); non-16/18 other hrHPV (*n* = 18); HPV non-targeted by NeuMoDx (*n* = 2); HPV negative (*n* = 2).

## Data Availability

The data presented in this study are available upon reasonable request from the corresponding author.

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
