# Peer review of "Clinical Validation of the Fully Automated NeuMoDx HPV Assay for Cervical Cancer Screening"

_viruses, 2022, doi:10.3390/v14050893_

Round 1
Reviewer 1 Report
This manuscript describes a new method (NeuMoDx HPV Assay) of PCR to determine HPV infection by measuring the E7 region in a large number of archived samples in order to validate this new technique using HC2, COBAS, and Alinity as referents.
The manuscript is interesting because try to find new strategies for HPV determination in cervical cancer screening.
The study has been well designed and the methodology is right.
It would be interesting to know the results in CIN 2 and CIN 3 separately
In the discussion and the conclusion, the authors define “NeuMoDx HPV Assay can be considered clinically validated for cervical cancer screening”. I think this statement is pretentious. These findings demonstrate the validity of the technique to detect HPV infection and the sensitivity in the detection of the cases CIN2+, with no inferiority with respect to the consolidated techniques, but it needs prospective studies in a real population to be considered as a valid technique in cervical cancer screening programs.
Reviewer 2 Report
The authors of this study present the clinical performance of a new HPV DNA test assay. The results of this study were nicely presented and easy to follow.
I have the following comments to the authors:
-In Introduction the authors mention that ‘To avoid the detection of too many transient infections resulting in over referral and over-treatment of women, HPV assays need to be clinically validated and comply with the international guidelines for HPV DNA test requirements to assure a high quality of primary HPV-based cervical screening’. One of the major drawbacks of the HPV DNA tests is their low specificity, thus a positive HPV DNA test should be triaged before referral to colposcopy. The authors present a new HPV DNA assay, and this sentence is confusing since transient HPV infections will be detected by HPV DNA tests. Do the authors imply that NeuMoDx HPV Assay will not detect transient HPV infections?
-In Introduction it is not clear that the HPV Assay that the authors present is DNA-based. Please make this clear from the start.
-The authors should present more information on the population of this study. Were these women attending for routine cervical cancer screening? Over what period and in what locations? When the authors report that CIN2+ was histologically confirmed, do they mean on punch biopsies, on cone specimens, or both? In this study did the authors include the last cervical sample before the histological diagnosis of CIN2+? What was the interval between the liquid-based cervical sample and the histological diagnosis? Were women with ASC-US excluded from the control group? How many women in the control group had HPV infection?
-The controls in this study are women with two negative cytology results. In other words, the ‘gold standard’ in this study is the cytology, which is known to have low sensitivity and specificity. There is a possibility that some women with two negative smears had CIN2+ that was missed by cytology. This misclassification will obviously influence the calculations for sensitivity and specificity. Did the authors consider this limitation in this study?
-In Methods authors do not provide the name of the statistical test they used.
-The reported specificity seems to be too high for an HPV DNA-based test. Authors should provide more details on the CIN2+ and control groups, in order to better understand what the population was this high specificity was calculated for.
-Do the authors expect that the performance of the assay will be affected by the type of the liquid medium (PreservCyt or SurePath)? Why did they perform the analyses separately according to the liquid medium? They should discuss this.
-The authors report that some samples did not give valid results. What do the authors mean by invalid results? That there was an error when running the sample?
-In Discussion authors should make a comparison between NeuMoDx HPV Assay and other HPV assays (including HPV mRNA-based tests). For example, they say that results are available within 60 minutes. How long do other HPV tests need?
Round 2
Reviewer 2 Report
I would like to thank the reviewers for their efforts to address all comments raised. I still have the following (minor) comments:
Comment 3:
A) Authors should add in the main text that their population is women attending national screening programme. Additionally, I think that it is still confusing which women were included in the reference population. In Methods, authors explain that reference population comprised of women with two normal smears, while in Results reference group is women with ≤CIN1. The following scenarios are unclear:
-If a woman with ASC-US or LSIL subsequently had two normal cytology smears, was she included in the reference population after the two normal smears?
-If a woman with ASC-US or LSIL underwent punch biopsy showing ≤CIN1, was she included in the reference population?
-If a woman with HSIL underwent punch biopsy showing ≤CIN1, was she included in the reference population?
-if a woman had abnormal cytology prior to study start date (e.g. LSIL on October 2009) and she was invited for repeat cytology after study start date (e.g. on April 2010), was she included in the study? Or did authors include only women whose last cytology smear prior to study start date was normal?
-If a woman had been treated for CIN prior to study start date and she was on post-treatment follow-up during the study duration, was she included in the study?
I suggest that the authors expound further on the inclusion criteria. These details are crucial and should be in the main manuscript, in order for the reader to understand the population the reported sensitivity and specificity were calculated for, without the need to read other citations.
B) In cytology-based screening programmes prior to HPV-based screening programmes, one of the uses for HVP testing was to triage women with ASC-US. If authors excluded women with ASC-US, do they think that this is a limitation of the study?
